# Towards a Transferable Acceleration Method for Density Functional Theory

**Zhe Liu, Yuyan Ni, Zhichen Pu, Qiming Sun, Siyuan Liu & Wen Yan**
Bytedance Seed
{qiming.sun, siyuan.liu}@bytedance.com

## Abstract

Recently, sophisticated deep learning-based approaches have been developed for generating efficient initial guesses to accelerate the convergence of density functional theory (DFT) calculations. While the actual initial guesses are often density matrices (DM), quantities that can convert into density matrices also qualify as alternative forms of initial guesses. Hence, existing works mostly rely on the prediction of the Hamiltonian matrix for obtaining high-quality initial guesses. However, the Hamiltonian matrix is both numerically difficult to predict and intrinsically non-transferable, hindering the application of such models in real scenarios. In light of this, we propose a method that constructs DFT initial guesses by predicting the electron density in a compact auxiliary basis representation using E(3)-equivariant neural networks. Trained *exclusively* on small molecules with up to 20 atoms, our model achieves an average 33.3% reduction in SCF iterations for molecules three times larger (up to 60 atoms). This result is particularly significant given that baseline Hamiltonian-based methods fail to generalize, often *increasing* the iteration count by over 80% or failing to converge entirely on these larger systems. Furthermore, we demonstrate that this acceleration is robustly scalable: the model successfully accelerates calculations for systems with up to 900 atoms (polymers and polypeptides) without retraining. To the best of our knowledge, this work represents the first and robust candidate for a universally transferable DFT acceleration method. We also released the SCFbench dataset and its accompanying code[1] to facilitate future research in this promising direction.

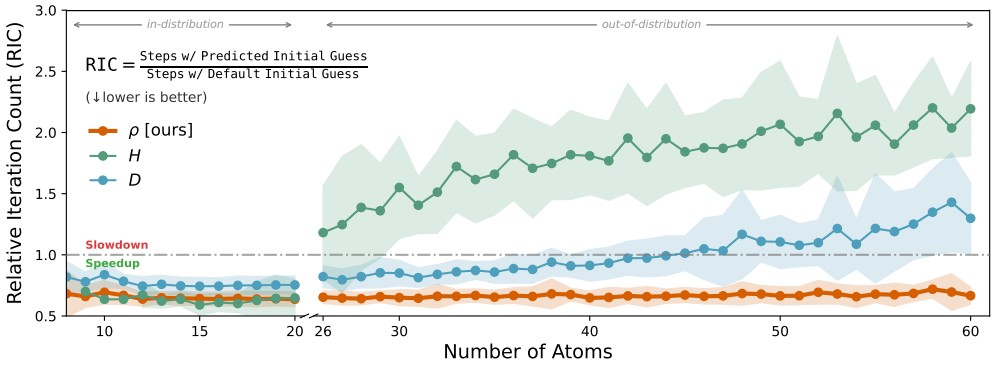

Figure 1: Comparison of deep learning-based DFT acceleration methods with different initial guess targets. The main metric, Relative Iteration Count (RIC), measures the ratio of SCF iterations required with a deep learning initial guess relative to a baseline. A smaller RIC means fewer SCF iterations required for convergence and is therefore preferable. While the three models perform similarly on in-distribution (ID) systems, on out-of-distribution (OOD) systems, our proposed method with electron density ($\rho$) as the target performs significantly better than methods based on Hamiltonian ($\boldsymbol{H}$) or density matrix ($\boldsymbol{D}$). More crucially, it shows a nearly constant scaling with increasing system size, which is an ideal property for the task of DFT acceleration.

---

[1]https://huggingface.co/datasets/ByteDance-Seed/SCFbench

# 1 INTRODUCTION

Density Functional Theory (DFT) (Hohenberg & Kohn, 1964; Kohn & Sham, 1965; Parr & Yang, 1994) is a cornerstone of computational chemistry, offering a powerful framework for predicting the electronic structure and properties of molecules. The most widely applied algorithm for solving the DFT problem is the self-consistent field (SCF) method, an iterative process that refines an initial guess for the density matrix until a converged solution is found. However, the iterative nature of SCF can be computationally expensive, particularly for large systems, creating a significant bottleneck in chemical discovery.

Machine learning (ML) offers a promising path to accelerate these calculations by providing a high-quality initial guess for the SCF procedure, as illustrated in Figure 2. A popular approach is to train models to predict the Hamiltonian matrix (Yu et al., 2023; 2024; Li et al., 2025c). However, this strategy faces critical limitations, particularly for the large molecules where acceleration is most needed. The poor performance stems from two distinct reasons. First, even when trained on datasets containing large molecules, the approach is hampered by numerical instability: small prediction errors in individual Hamiltonian matrix elements can be magnified into large, physically nonsensical errors for the system as a whole (Li et al., 2025c). Second, and more critically, the approach fails to scale to molecules larger than those seen during training.

This lack of transferability is rooted in a fundamental limitation of the theory itself: the core ansatz of Kohn-Sham DFT is that a real system of interacting electrons can be represented by a fictitious, non-interacting system that shares the *exact same electron density* (Kohn & Sham, 1965). This makes the electron density the fundamental physical observable, rather than the Hamiltonian matrix. A key consequence is that the electron density associated with a specific chemical environment is highly transferable. The Hamiltonian matrix, however, does not share this property; it contains matrix elements for every pair of atoms in a molecule, regardless of the distance separating them, making its prediction sensitive to the molecule's entire global structure. This makes the Hamiltonian a difficult target for extrapolating to larger, more complex chemical environments.

As an alternative, predicting density matrices for generating DFT initial guesses has been proposed (Shao et al., 2023; Hazra et al., 2024; Febrer et al., 2025). However, this strategy is strongly basis-set dependent. In particular, when diffuse functions are included, density matrix elements span a much larger numerical range, amplifying numerical uncertainties.

We argue that a more fundamental and transferable target for prediction is the electron density itself. Previous works have attempted to predict the electron density on real-space grids (Brockherde et al., 2017). However, such grid-based predictions are not directly suitable for constructing an SCF initial guess, as most DFT functionals require not only the density but also its gradients, which are not readily available from grid-based predictions alone. Furthermore, even when similar ideas were

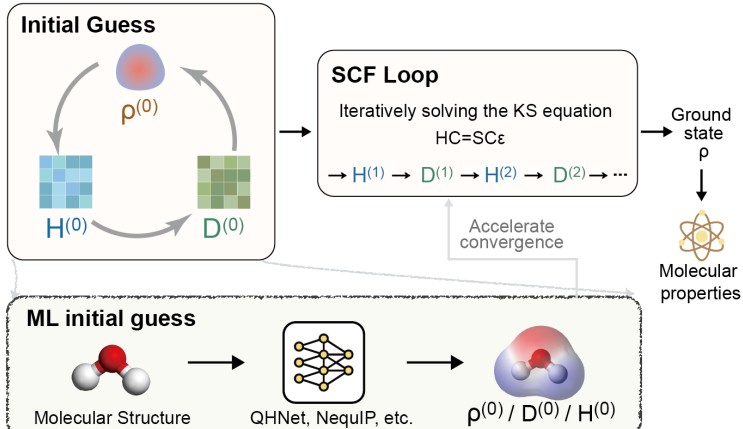

Figure 2: Top left: The Hamiltonian ($H$), density matrix ($D$), and electron density ($\rho$) are interdependent, so any of them can serve as an initial guess. Top center: An SCF loop iteratively finds the ground state from the given initial guess. Bottom: An ML model predicts an initial guess from a molecular structure to accelerate the SCF loop.

proposed (Grisafi et al., 2019; Fu et al., 2024), a practical method for using the predicted density to accelerate DFT calculations was never fully realized.

We propose a new paradigm that overcomes these limitations. We train a model to predict the expansion coefficients of the electron density in a compact auxiliary basis and, crucially, demonstrate how to use this prediction to construct a transferable initial guess for the SCF process. Our most significant finding is that the electron density is a *highly transferable and scalable property*. A model trained on molecules with only 20 atoms can be applied directly to systems with 60 atoms or even 900 atoms without fine-tuning. On benchmark tests involving all system sizes, our method achieves an substantial improvement over the Hamiltonian- and density-matrix-based approaches (illustrated in Figure 1).

This electron-density-centric approach offers several additional advantages. First, from a practical standpoint, processing density coefficients is significantly more efficient. The number of coefficients in an auxiliary basis scales linearly with system size, whereas the Hamiltonian and density matrices scale quadratically. Second, the electron density can possess lower symmetry ($L$) than the Hamiltonian, which is particularly beneficial for equivariant neural networks where the computational complexity of the tensor product scales as $O(L^6)$. Finally, the local nature of electron density makes our approach highly data-efficient, requiring a smaller training set to achieve high accuracy.

To facilitate further research in this direction, we introduce a new dataset SCFbench containing the electron densities of molecules composed of up to seven different elements. We provide benchmark results for two prominent ML architectures, demonstrating how our electron density prediction task can be seamlessly integrated into existing models to accelerate quantum chemical calculations.

The main contributions of this work can be summarized as follows:

- We propose a new paradigm for DFT acceleration that targets the electron density—a more fundamental, local, and data-efficient quantity—to provide a high-quality initial guess for SCF calculations. Specifically, we take the efforts to implement the *procedure for converting the electron density into an initial guess*, the absence of which was the direct cause of the under-development of this principled paradigm.
- We introduce SCFbench, the first public dataset of electron density coefficients specifically designed for developing and benchmarking DFT acceleration methods.
- We systematically benchmark our electron-density-centric approach for both in-domain and transferring settings on SCFbench. Results indicate that our approach shows remarkable transferability not only to larger molecules but also across different exchange-correlation (XC) functionals and orbital basis sets.

## 2 BACKGROUND

### 2.1 KOHN-SHAM DFT AND THE SELF-CONSISTENT FIELD METHOD

Kohn-Sham (KS) DFT provides a systematic framework to construct the energy functional $E[\rho(\boldsymbol{r})]$ of a system based on its electron density (Parr & Yang, 1994). The electron density $\rho(\boldsymbol{r})$ in KS-DFT is constructed from the density matrix $\boldsymbol{D}$ and a set of basis functions $\{\phi_\mu(\boldsymbol{r})\}$:

$$\rho(\boldsymbol{r}) = \sum_{\mu,\nu} \boldsymbol{D}_{\mu\nu}\phi_\mu(\boldsymbol{r})\phi_\nu(\boldsymbol{r}). \tag{1}$$

The density matrix is derived from the molecular orbital coefficients $\boldsymbol{C}$

$$D_{\mu\nu} = \sum_i C_{\mu i}C_{\nu i}. \tag{2}$$

Minimization the total energy with respect to the orbital coefficients leads to a generalized eigenvalue equation:

$$\boldsymbol{H}[\rho]\boldsymbol{C} = \boldsymbol{S}\boldsymbol{C}\epsilon. \tag{3}$$

Here, $\boldsymbol{S}$ is the overlap matrix for the non-orthogonal basis functions, and $\epsilon$ is the diagonal matrix of orbital energies. The Kohn-Sham Hamiltonian matrix $\boldsymbol{H}$ is an effective single-particle Hamiltonian. $\boldsymbol{H}$ is composed of three distinct terms:

$$\boldsymbol{H} = \boldsymbol{H}_{\mathrm{core}} + \boldsymbol{J} + \boldsymbol{V}_{\mathrm{xc}}. \tag{4}$$

The core Hamiltonian ($H_{\mathrm{core}}$) is determined solely by the molecular geometry and basis set. The remaining terms capture the electronic interactions: the Coulomb matrix ($J$) for classical electron repulsion and the XC matrix ($V_{\mathrm{xc}}$) for quantum mechanical effects.

A significant computational challenge arises from the fact that both $J$ and $V_{\mathrm{xc}}$ depend on the density matrix $D$, which in turn is constructed from the orbital coefficients $C$. This interdependence necessitates an iterative procedure known as the SCF method (Szabo & Ostlund, 1996). One common approach to solving the SCF problem is to begin with an initial guess for the density matrix, $D$. From $D$, an initial Hamiltonian $H$ is constructed. Solving the eigenvalue problem yields new orbital coefficients $C'$, which are used to compute an updated density matrix, $D'$. This cycle, $D \rightarrow H \rightarrow C' \rightarrow D'$, is repeated until the density matrix converges and the solution is deemed self-consistent. A common strategy for this initial guess, such as the default `minao` method in PySCF (Sun et al., 2020), is the superposition of atomic densities (SAD) (Lehtola, 2019; Van Lenthe et al., 2006).

## 2.2 Constructing the Kohn-Sham Matrix from Predicted Density

The key insight for our work lies in how the electronic terms are constructed from the predicted electron density.

Using the density fitting (Dunlap, 2000) approximation, the electron density $\rho(r)$ can be expanded in terms of an auxiliary basis set $\{\chi_k(r)\}$ with the expansion coefficients $c_k$:

$$\rho(r) \approx \tilde{\rho}(r) = \sum_k c_k \chi_k(r). \tag{5}$$

These auxiliary basis functions are atom-centered. Typical auxiliary basis sets include def2-universal-jfit (Weigend, 2006) and the even-tempered basis (ETB) (Bardo & Ruedenberg, 1974), parameterized by $\beta$. A smaller $\beta$ yields a larger ETB basis. The size of the auxiliary functions is typically three to five times that of the atomic orbital basis functions, which is significantly smaller than the number of orbital pairs in $H$ and $D$. In our approach, the *auxiliary coefficients* $c_k$ are the primary quantities predicted using a machine learning model.

With the auxiliary basis expansion, both the electron density and its gradient can be directly evaluated. This allows us to efficiently evaluate the XC matrix for generalized gradient approximation (GGA) functionals. Additionally, the Coulomb matrix $J$, while formally dependent on the density matrix $D$, can be computed efficiently from the coefficients $\{c_k\}$ using the density fitting approximation.

This feature makes the GGA framework particularly well-suited for our approach, as a machine learning prediction of the density coefficients $\{c_k\}$ is sufficient to assemble the entire Kohn-Sham Hamiltonian matrix $H$. With additional approximations, extensions to more complex functional types are possible. Explicit formulas for constructing $J$ and $V_{\mathrm{xc}}$ for general XC functionals from the auxiliary density are provided in Appendix E.

Compared to computing $H$ directly from the full density matrix $D$, our approach introduces an approximation to $J$ and $V_{\mathrm{xc}}$ via the fitted density $\tilde{\rho}(r)$. However, The error in this approximation can be systematically reduced by increasing the number of auxiliary basis functions used in the expansion (see Appendix C for an illustrative example).

## 2.3 Equivariant Neural Networks

Physical properties of molecular systems are inherently independent of the choice of coordinate system. Under spatial transformations such as rotations, translations, or reflections, quantities like energy and electron density should transform accordingly, preserving their physical meaning. E(3)-equivariant neural networks are specifically designed to respect these symmetries, where E(3) denotes the Euclidean group of all such transformations (Kondor & Trivedi, 2018; Geiger & Smidt, 2022).

Formally, an equivariant model $\Phi$ satisfies the following property: when the input atomic coordinates $\{r_i\}$ are transformed by an operation $g \in E(3)$, the output $O$ transforms according to a corresponding representation $\mathcal{D}(g)$,

$$\Phi(g \cdot \{r_i\}) = \mathcal{D}(g)\Phi(\{r_i\}) \tag{6}$$

Here, $\mathcal{D}(g)$ is the appropriate representation for the output type. For scalar quantities such as total energy, $\mathcal{D}(g)$ is the identity, reflecting invariance under transformation. For tensorial properties, such as electron density coefficients in a spherical harmonics basis, $\mathcal{D}(g)$ corresponds to the Wigner D-matrix, which encodes the rotation of these higher-order objects. Incorporating such symmetry constraints into the network architecture via tensor product operations provides a strong inductive bias, improving generalization and data efficiency for molecular property prediction.

## 3   RELATED WORK

### 3.1   HAMILTONIAN PREDICTION

Recent works have developed neural networks for direct prediction of the Kohn-Sham Hamiltonian. PhiSNet (Unke et al., 2021) uses SE(3)-equivariant layers to reconstruct molecular wavefunctions and densities. QHNet (Yu et al., 2023) introduces an efficient SE(3)-equivariant graph network for Hamiltonian prediction with reduced tensor operations. SPHNet (Luo et al., 2025) incorporates adaptive sparsity into equivariant networks. QHFlow (Kim et al., 2025) employs high-order equivariant flow matching to generate Hamiltonians conditioned on molecular geometry. The most scalable Hamiltonian model to date is proposed by Li et al. (2025c), which introduces the Wavefunction Alignment Loss (WALoss) to enable Hamiltonian prediction for large molecules and significantly improve the derived energy compared to previous Hamiltonian models. However, its accuracy for energy prediction remains much lower than that of direct energy models.

Other related Hamiltonian prediction works include Li et al. (2022); Zhang et al. (2024); Tang et al. (2024).

### 3.2   DENSITY MATRIX PREDICTION

Recent studies have begun to explore direct prediction of the density matrix (Shao et al., 2023; Hazra et al., 2024; Febrer et al., 2025). The first two works applied kernel-based methods to a small set of molecules, but not yet leveraging equivariant neural networks. The third used a small numerical atomic orbitals basis set and focused on small molecules. While these approaches have advanced the field, density matrix prediction still faces challenges with transferability and scalability: the density matrix elements are highly sensitive to the choice of basis set, which can limit generalization across chemical systems.

### 3.3   ELECTRON DENSITY PREDICTION

ML prediction of electron density has been widely studied (Brockherde et al., 2017; Ellis et al., 2021; Jørgensen & Bhowmik, 2022; Focassio et al., 2023; Rackers et al., 2023; Lee & Kim, 2024; Voss, 2024; Elsborg et al., 2025; Li et al., 2025a). Early efforts typically represented the density on real-space grids, which introduced redundancy and high computational cost. In contrast, representing the density with one-center auxiliary functions provides a more efficient representation while maintaining good accuracy (Grisafi et al., 2019). Fu et al. (2024) introduced an accurate and efficient model, SCDP, for predicting electron density on real-space grids using even-tempered Gaussian functions as auxiliary basis sets, augmented with off-center virtual orbitals. However, due to the common lack of support of using electron density as an initial guess in quantum chemistry software, none of these works have explored how the predicted density could be leveraged to accelerate DFT calculations.

### 3.4   PUBLIC HAMILTONIAN DATASETS

Several publicly available datasets provide Hamiltonian matrices for molecular systems and are closely related to this work. MD17 (Schütt et al., 2019) contains Hamiltonians for thousands of structures of four small molecules, computed with the def2-SVP basis set and PBE functional. QH9 (Yu et al., 2024) extends the QM9 (Ramakrishnan et al., 2014) dataset with over 130,000 stable geometries and molecular dynamics trajectories, providing precise Hamiltonians and open-source benchmarks for model development. The nablaDFT (Khrabrov et al., 2022) and its extension, $\nabla^2$DFT (Khrabrov et al., 2024), offer a large collection of drug-like molecules with millions of

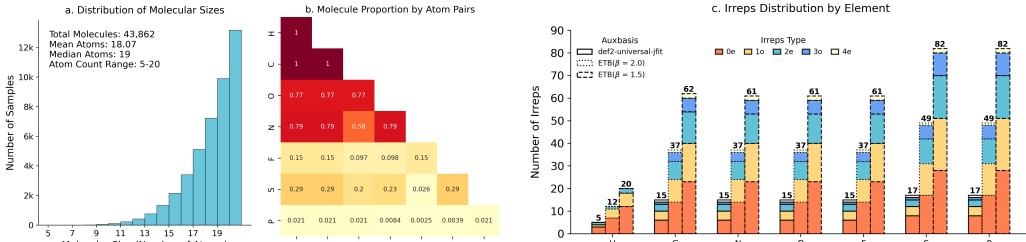

Figure 3: Statistical analysis of the SCFbench dataset. (a) Distribution of molecule sizes, (b) Proportion of molecules containing each individual element (H, C, N, O, F, P, S) and element pair present in the dataset, and (c) decomposition of electron density into irreducible representations (irreps) over the auxiliary basis sets.

conformations and associated quantum chemistry properties, including Hamiltonians and geometry optimization trajectories. Other related datasets include QCML (Ganscha et al., 2025), which uses numerical atomic orbitals, and PubChemQH (Li et al., 2025c), which is not yet publicly released.

## 4 THE SCFBENCH DATASET

To support research on scalable and transferable initial guess methods, we introduce the SCFbench dataset. It was constructed by applying a fragmentation procedure, similar to that of Zheng et al. (2025), to drug-like molecules within the ChEMBL database (Zdrazil et al., 2024). A subset of fragments containing 20 atoms or less was selected, covering the elements H, C, N, O, F, P, and S. SCFbench dataset has several features designed to evaluate transferability and scalability.

In addition to Hamiltonians and density matrices, SCFbench provides electron density expansion coefficients for three distinct auxiliary basis sets: the computationally efficient def2-universal-jfit (Weigend, 2006), and two even-tempered basis (ETB) sets built from def2-SVP with $\beta = 2.0$ and $\beta = 1.5$ (Bardo & Ruedenberg, 1974). The resulting dataset of 43,862 molecules is randomly split into training, validation, and test sets with a ratio of 8:1:1. The test set is used for evaluating the in-distribution performance of a model, so it is also called as the in-distribution (ID) test set. This dataset is well-suited for the electron density prediction task and is designed to be lightweight and easily extensible.

A key feature of the dataset is its dedicated out-of-distribution (OOD) test set, designed to address the challenge of system size transferability. This set comprises 1,050 molecules, consisting of 30 molecules for each atom count from 26 to 60, allowing for the evaluation of models on systems significantly larger than the training data. The OOD set also includes the number of SCF cycles required for each molecule.

The data was generated using the PBE functional (Perdew et al., 1996), a pure GGA functional. This choice was made because, as discussed in section 2, the Kohn-Sham Hamiltonian for GGA functionals can be constructed directly from the electron density coefficients, making it a suitable framework for developing and testing our proposed method. It is also worth noting that while SCF convergence is often more challenging for GGA functionals than for hybrids (Mori-Sánchez et al., 2008; Rabuck & Scuseria, 1999), they remain underrepresented in existing Hamiltonian datasets.

For the DFT calculations, we used the def2-SVP basis set (Weigend & Ahlrichs, 2005), a [99, 590] atom grid, and an SCF energy convergence tolerance of $1 \times 10^{-10}$.

## 5 METHODS

### 5.1 EVALUATION

The primary goal of our evaluation is to assess the practical value of using ML-predicted electron density to accelerate SCF calculations. We evaluate the models on their ability to accelerate SCF convergence for molecules both within the training distribution (ID) and for molecules significantly

larger than those seen during training (OOD). This directly tests the crucial property of size transferability. Furthermore, we assess the robustness of the models by testing their transferability across different XC functionals and atomic orbital basis sets.

Our main metric is the *Relative Iteration Count* (RIC), defined as the number of SCF cycles required for convergence using the ML-predicted initial guess, normalized by the number of cycles required using the standard SAD initialization (`minao`). A lower RIC means greater acceleration.

In addition, we define a calculation as converged if it reaches the required PySCF default tolerance within 50 iterations. This allows us to report a *convergence rate*, which is a crucial metric for practical usability. We use the default PySCF DIIS settings (space size of 8), no level shifting, and no damping. For the ID vs. OOD comparison, we evaluate on a random subset containing 1% of the test set and the full OOD test set, respectively. A comprehensive set of additional metrics is provided in Appendix H.

While the RIC provides a hardware-independent and robust measure of initial guess quality, it might be interesting to see how much acceleration the models can actually bring to the wall time. To this end, a detailed wall-time analysis is provided in Appendix G.

## 5.2 MODEL ARCHITECTURES

Instead of designing a new architecture from the ground up, we adapt two classical models—NequIP (Batzner et al., 2022) and QHNet (Yu et al., 2023)—by modifying their final prediction heads.

**NequIP**  NequIP (Batzner et al., 2022) is an E(3)-equivariant graph neural network that represents atomistic systems as graphs. Its core operation is a symmetry-preserving convolution where messages, constructed via a tensor product of neighbor features and a filter made of learnable radial functions and spherical harmonics, are passed between atoms. This process iteratively refines each atom's features, which are geometric tensors (irreducible representations) of varying orders ($l$). Originally, NequIP's architecture concluded with a simple head that processed scalar ($l = 0$) features to predict atomic energies.

**QHNet**  QHNet (Yu et al., 2023) is an efficient SE(3)-equivariant model designed to predict quantum tensors. Its architecture is distinguished by node-wise interaction layers that use an attention-like mechanism and a Norm Gate that dynamically rescales higher-order tensor features. QHNet was originally designed with a large, multi-stage prediction head that used a Tensor Expansion module to construct the final Hamiltonian matrix from pair-wise atomic features.

**Species-dependent Equivariant Prediction Head**  We replace the original prediction heads of both models with a single, species-dependent equivariant linear layer. This simple layer directly maps the final node features from the backbone to a new output feature, $h_{\text{out}}^i$, which are the density coefficients containing irreducible representations from order $l = 0$ to $l = 4$. The layer's weights are conditioned on the atomic species, allowing the model to learn a distinct final mapping for each chemical element. For NequIP, this modification has a negligible impact on the total parameter count. For QHNet, however, replacing its complex original head results in a significant efficiency gain, with the final model retaining only about one-quarter of the original parameters (see Table 1).

## 5.3 TRAINING PROCEDURE

All density coefficients models were trained by minimizing a composite loss function, $\mathcal{L}$, calculated per atom. This loss is the sum of the mean absolute error (MAE) and the root mean square error (RMSE) of the coefficients, averaged over all atoms in the batch:

$$\mathcal{L} = \left( \frac{1}{A} \sum_{a=1}^{A} \frac{1}{N_a} \sum_{i=1}^{N_a} |\hat{c}_{a,i} - c_{a,i}| \right) + \sqrt{\frac{1}{A} \sum_{a=1}^{A} \frac{1}{N_a} \sum_{i=1}^{N_a} (\hat{c}_{a,i} - c_{a,i})^2} \tag{7}$$

where $A$ is the total number of atoms, $N_a$ is the number of coefficients for atom $a$, and $\hat{c}_{a,i}$ and $c_{a,i}$ are the predicted and ground-truth coefficients. The ground-truth $c_{a,i}$ are derived from the

final, converged electron density of the DFT calculation. Other training details are described in Appendix F.

## 6 EXPERIMENTS

In this section, we present benchmark results for the SCFbench dataset using the modified NequIP and QHNet models, focusing on their performance in accelerating SCF calculations. The results are summarized in Table 1 and illustrated in Figure 1.

### 6.1 SCF ACCELERATION WITH PREDICTED DENSITY (ID AND OOD)

Table 1: Results on the SCFbench benchmark dataset. The best results for each dataset are highlighted in bold and the second bests are underlined. Best settings for each prediction target are marked in  gray . Results for an extended set of metrics are available in Appendix H.

| Prediction Target | Model | # Param. | ID Test | | OOD Test | |
|---|---|---|---|---|---|---|
| | | | Convergence ↑ | RIC ↓ | Convergence ↑ | RIC ↓ |
| **Hamiltonian** | Ground Truth | - | 100% | 29.22% | 100% | 26.96% |
| | QHNet | 20.5M | 100% | 63.20% | 97.43% | 179.47% |
| **Density Matrix** | Ground Truth | - | 100% | 27.57% | 100% | 26.62% |
| | QHNet | 20.5M | 100% | 70.45% | 99.71% | 91.69% |
| **Density Coefficients** def2-universal-jfit | Ground Truth | - | 100% | 62.80% | 100% | 60.45% |
| | QHNet | 5.9M | 100% | 66.90% | 100% | 73.26% |
| | NequIP-S | 2.7M | 100% | 74.90% | 100% | 89.46% |
| | NequIP-M | 36.9M | 100% | 64.82% | 100% | 69.10% |
| | NequIP-L | 50.0M | 100% | 63.78% | 100% | **66.68%** |
| **Density Coefficients** ETB, $\beta = 2.0$ | Ground Truth | - | 100% | 58.96% | 100% | 55.05% |
| | QHNet | 5.9M | 100% | 68.62% | 100% | 79.36% |
| | NequIP-S | 2.7M | 100% | 82.20% | 100% | 93.28% |
| | NequIP-M | 36.9M | 100% | 67.31% | 100% | 78.39% |
| | NequIP-L | 50.0M | 100% | **62.48%** | 100% | 70.42% |
| **Density Coefficients** ETB, $\beta = 1.5$ | Ground Truth | - | 100% | 43.26% | 100% | 39.66% |
| | QHNet | 5.9M | 100% | 78.05% | 100% | 82.76% |
| | NequIP-S | 2.7M | 100% | 89.80% | 99.24% | 127.16% |
| | NequIP-M | 36.9M | 100% | 76.82% | 99.62% | 108.17% |
| | NequIP-L | 50.0M | 100% | 68.85% | 99.90% | 81.33% |

The Hamiltonian prediction model exhibits significant limitations. While it achieves a low RIC on the in-distribution test set (63.20%), which is comparable to performance reported in other works (Yu et al., 2024), its performance collapses on the OOD test set. The relative iteration count increases to 179.47%. More alarmingly, the model suffers from a non-convergence problem, failing to converge for over 2.5% of the OOD molecules. Unlike chemically-grounded methods like SAD, the ML model can produce unphysical initial guesses, especially for larger molecules, leading to a failure of the SCF procedure.

Predicting the density matrix offers an improvement over the Hamiltonian but still falls short in transferability. It achieves a solid RIC of 70.45% relative iteration count on the ID set and maintains a high convergence rate on the OOD set. However, its performance degrades on larger molecules, with the RIC increasing to 91.69% for the OOD set. As illustrated in Figure 1, its performance clearly worsens as system size increases, highlighting that the density matrix remains a challenging target for size transferability.

In stark contrast, our density-based models demonstrate excellent scalability. On the ID test set, the best models (NequIP-L) achieve RICs of 62-64%, nearing the theoretical limit imposed by the ground truth density. Critically, this strong performance is maintained on the OOD test set. For the def2-universal-jfit basis, the NequIP-L model's acceleration is remarkably consistent, with an RIC of 63.78% on the ID set and 66.68% on the OOD set, and it achieves a 100% convergence rate across all tests. This remarkable consistency proves that electron density is a highly transferable

property, enabling models trained on small molecules to effectively accelerate calculations for much larger ones. Further emphasizing this point, when we task the QHNet electron density model with predicting electron density instead of the Hamiltonian matrix, its RIC improves dramatically to 73.26% on the OOD set, highlighting that the choice of a transferable physical quantity is more critical than the specific model architecture.

Analyzing the ground truth results reveals the theoretical limits of this approach. The ground truth Hamiltonian and Density Matrix provide the best possible initial guess, requiring only one SCF cycle in theory; the remaining 27-29% RICs are attributed to numerical precision differences. For density coefficients, the potential acceleration depends on the expressiveness of the auxiliary basis, with larger bases like ETB ($\beta = 1.5$) offering greater potential for acceleration (a theoretical limit of ~40%) than the compact def2-universal-jfit basis (~60%). Our ML models come very close to reaching this RIC limit for the def2-universal-jfit basis, demonstrating the learnability of the task. The performance gap for the larger ETB basis sets, however, highlights a promising avenue for future work. Improved model architectures could potentially capture more information from these expressive bases, pushing the acceleration even closer to the theoretical limit.

## 6.2 SCALING TO LARGE-SCALE SYSTEMS

Table 2: Scalability test on the QMugs dataset (100–200 atoms). For density coefficients, we use NequIP-L trained with the def2-universal-jfit basis. Convergence indicates the percentage of molecules that converged within 50 iterations. The average RIC is reported for converged molecules only.

| # Atoms | Density Coefficients | | Hamiltonian | | Density Matrix | |
|---|---|---|---|---|---|---|
| | RIC ↓ | Convergence ↑ | RIC ↓ | Convergence ↑ | RIC ↓ | Convergence ↑ |
| 100 | 75.36% | 100% | 224.11% | 20% | 190.19% | 50% |
| 110 | 78.64% | 100% | 286.00% | 30% | 268.75% | 10% |
| 120 | 73.42% | 100% | 281.62% | 10% | 233.33% | 10% |
| 130 | 78.10% | 100% | - | 0% | 306.67% | 10% |
| 140 | 75.61% | 100% | - | 0% | 326.67% | 10% |
| 150 | 77.12% | 100% | - | 0% | - | 0% |
| 160 | 79.07% | 100% | - | 0% | - | 0% |
| 170 | 80.87% | 100% | - | 0% | - | 0% |
| 180 | 76.70% | 100% | - | 0% | - | 0% |
| 190 | 81.77% | 100% | - | 0% | - | 0% |
| 200 | 77.34% | 100% | - | 0% | - | 0% |

To further evaluate the scalability of our method beyond the SCFbench OOD test set, we conducted additional experiments on the QMugs dataset (Isert et al., 2022), selecting a total of 110 molecules ranging from 100 to 200 atoms. As shown in Table 2, our density-based method maintains a consistent RIC between 0.73 and 0.82 with a 100% convergence rate up to 200 atoms. In contrast, Hamiltonian and density matrix prediction methods exhibit severe degradation, with convergence rates dropping to near zero for systems larger than 120 atoms due to poor initial guess quality leading to SCF divergence.

Furthermore, we evaluated two large-scale cases: a Glycine-100 polypeptide (703 atoms) and a Polypropylene polymer chain ($H[CH_2(CH_3)CH]_{100}CH_3$, 905 atoms):

- **Glycine-100:** Converged in 10 iterations (vs. 17 for `minao`).

- **Polypropylene:** Converged in 8 iterations (vs. 12 for `minao`).

Our method successfully accelerated convergence in both cases. However, both Hamiltonian and density matrix methods failed with out-of-memory errors. This highlights a critical advantage of our approach: predicting density coefficients is a node-wise task. In contrast, predicting Hamiltonian or density matrices is an edge-wise task, requiring the construction of large $N \times N$ matrices.

Table 3: Transferability of NequIP-L model across different functionals and basis sets. The model is trained on PBE/def2-SVP and evaluated on various settings. RICs are reported for both ID and OOD sets.

| Functional (Family: Name) | Basis Set | RIC (ID Test)↓ | RIC (OOD Test)↓ |
|---|---|---|---|
| ***In-distribution setting*** | | | |
| GGA: PBE | def2-SVP | 63.78% | 66.68% |
| ***Transferring to different XC functionals*** | | | |
| GGA: BLYP | | 71.38% | 71.22% |
| meta-GGA: SCAN | def2-SVP | 88.15% | 86.45% |
| Hybrid GGA: B3LYP5 | | 84.63% | 83.72% |
| Hybrid GGA: PBE0 | | 85.99% | 85.51% |
| ***Transferring to different atomic orbital basis sets*** | | | |
| | def2-TZVP | 76.68% | 75.24% |
| GGA: PBE | def2-TZVPPD | 77.07% | 75.81% |
| | def2-QZVP | 77.81% | 75.98% |
| ***Transferring to different XC functionals AND basis sets*** | | | |
| Hybrid GGA: B3LYP5 | def2-TZVP | 87.70% | 85.47% |

## 6.3 FUNCTIONAL AND BASIS SET TRANSFERABILITY

A key advantage of targeting electron density is its theoretical independence from the specific XC functional and orbital basis set used in a calculation (Kohn & Sham, 1965). To test this in practice, we evaluate the transferability of a single model—the NequIP-L model trained on PBE/def2-SVP with the def2-universal-jfit auxiliary basis—across a range of different functionals and larger orbital basis sets.

For meta-GGA and hybrid functionals, constructing the initial Fock matrix from the predicted electron density requires specific approximations for the kinetic energy density (meta-GGA) or the Hartree-Fock exchange term (hybrid). We detail these treatments in Appendix E. Despite these necessary approximations, Table 3 shows the practical robustness of the density-based approach. While performance moderately degrades compared to the original PBE/def2-SVP setting, the model still provides meaningful acceleration, particularly for the OOD set, showcasing its utility in diverse computational chemistry workflows.

Notably, our model achieves an RIC of 85.47% (OOD) on B3LYP5/def2-TZVP, where B3LYP5 refers to the B3LYP hybrid functional (Lee, 1988) with the VWN5 correlation component (Vosko et al., 1980)—matching the setup used in Li et al. (2025c). Despite training on a dataset consisting of much smaller molecules and with different functional and basis set choices, our density-based approach delivers comparable acceleration performance for molecules of similar size. This highlights the strong transferability and data efficiency of electron density prediction, even when evaluated under conditions aligned with state-of-the-art Hamiltonian-based models.

## 7 CONCLUSION

By targeting the electron density, this work provides a practical and reliable solution to the long-standing challenge of creating a scalable initial guess for SCF calculations. We have shown that a single model, trained on a modest dataset of small molecules, can serve as a "drop-in" accelerator for a wide range of systems, including those significantly larger than the training data, and across various functionals and basis sets. The robustness of our method marks a steady step towards a universally applicable tool for the computational chemistry community.

To facilitate further progress, we have released the SCFbench dataset, a comprehensive benchmark designed to test these crucial aspects of transferability and scalability. Future work can build on this foundation in several key directions. While our models approach the theoretical performance limit for compact auxiliary basis sets, a gap remains for more expressive bases; developing more powerful neural network architectures could close this gap and unlock even greater acceleration. Furthermore, extending the SCFbench dataset to include a wider range of the periodic table and periodic systems will be vital for pushing this promising method towards true universality.

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

## A    THE ROLE OF THE SCF PROCEDURE

While our results demonstrate that machine learning models can predict the electron density with high accuracy, which are often yielding energies within chemical accuracy (see Appendix J), we retain the SCF procedure to ensure the physical consistency of the full electronic structure. Direct prediction models are highly effective for properties like total energy; however, many downstream tasks in computational chemistry, such as calculating NMR shielding tensors, dipole moments, or excited states, require the explicit, self-consistent Kohn-Sham orbitals and eigenvalues. By using the predicted density to initialize and accelerate the SCF loop rather than replace it, we ensure that the resulting wavefunction obeys the variational principle and provides a unified, *ab initio* basis for deriving all ground-state properties, rather than relying on separate regressors for each observable.

## B    DENSITY CONSTRAINTS AND NORMALIZATION

A theoretical requirement for the electron density is that it integrates to the total number of electrons ($N_e$) and remains non-negative. Our model predicts expansion coefficients $\{c_k\}$ in an auxiliary basis, which are not strictly constrained to satisfy $\int \rho(\boldsymbol{r})d\boldsymbol{r} = N_e$ during the ML inference.

However, explicit normalization of the predicted density is not necessary for minimizing RIC, and in our experiments, enforcing it explicitly even slightly degraded performance (OOD RIC changes from 66.68% to 67.81%). This is for two reasons:

1. Standard initial guesses (e.g., SAD/`minao`) are often constructed from superpositions of spherical atoms and do not strictly integrate to the correct $N_e$ before the first cycle.

2. The predicted coefficients are used solely to construct the initial Fock matrix. When this matrix is diagonalized, we select the lowest $N_e/2$ orbitals (for restricted DFT) to construct the new density matrix. This eigensolver step inherently enforces the correct number of electrons for the subsequent iteration.

## C    ERROR SOURCES IN CONSTRUCTING HAMILTONIAN FROM DENSITY COEFFICIENTS

Table 4 compares the number of SCF cycles required for convergence for the D-Glucose molecule ($C_6H_{12}O_6$) using different sets of auxiliary basis sets and their corresponding number of basis functions. As shown, increasing the basis set size can reduce the number of SCF cycles to as low as 38.5% of the baseline.

Table 4: Effect of auxiliary basis set size on SCF convergence for D-Glucose ($C_6H_{12}O_6$). SCF iteration ratios are reported as the number of iterations required for convergence, normalized to the default `minao` initial guess. With the def2-SVP basis set, D-Glucose has 228 basis functions. All results are obtained using the ground truth density coefficients as the initial guess.

| Auxiliary Basis Set | Number of Basis Functions | SCF Cycles | SCF Iter. Ratio (%) |
|---|---|---|---|
| def2-universal-jfit | 720 | 7 | 53.8 |
| ETB ($\beta = 2.0$) | 1740 | 7 | 53.8 |
| ETB ($\beta = 1.5$) | 2898 | 5 | 38.5 |

## D    COMPUTING THE DENSITY COEFFICIENTS

In our work, the machine learning target is the set of expansion coefficients $\{c_k\}$ that represent the electron density $\rho(\boldsymbol{r})$ in a given auxiliary basis set $\{\chi_k(\boldsymbol{r})\}$. There are at least two principled ways to determine these ground-truth coefficients from a converged DFT calculation.

The first approach is to minimize the squared error of the density itself, which corresponds to an L2 projection of the density onto the auxiliary basis. The objective is to find the coefficients $\{c_k\}$ that

solve the following minimization problem:

$$\min_{\{c_k\}} \int \left| \rho(\boldsymbol{r}) - \sum_k c_k \chi_k(\boldsymbol{r}) \right|^2 d\boldsymbol{r}. \tag{8}$$

This leads to a system of linear equations:

$$\sum_l S_{kl}^{\text{aux}} \boldsymbol{c}_l = \int \rho(\boldsymbol{r}) \chi_k(\boldsymbol{r}) d\boldsymbol{r}, \tag{9}$$

where $S_{kl}^{\text{aux}} = \int \chi_k(\boldsymbol{r})\chi_l(\boldsymbol{r})d\boldsymbol{r}$ is the overlap matrix of the auxiliary basis functions.

The second approach, which is the standard method in density fitting, is to minimize the error in the Coulomb repulsion energy. The objective is to minimize the self-repulsion of the residual density:

$$\min_{\{c_k\}} \iint \frac{\left(\rho(\boldsymbol{r}) - \sum_k \boldsymbol{c}_k \chi_k(\boldsymbol{r})\right)\left(\rho(\boldsymbol{r}') - \sum_l \boldsymbol{c}_l \chi_l(\boldsymbol{r}')\right)}{|\boldsymbol{r} - \boldsymbol{r}'|} d\boldsymbol{r} d\boldsymbol{r}'. \tag{10}$$

This leads to a different system of linear equations:

$$\sum_k \left( \iint \frac{\chi_k(\boldsymbol{r})\chi_k(\boldsymbol{r}')}{|\boldsymbol{r} - \boldsymbol{r}'|} d\boldsymbol{r} d\boldsymbol{r}' \right) v_k = \iint \frac{\rho(\boldsymbol{r})\chi_l(\boldsymbol{r}')}{|\boldsymbol{r} - \boldsymbol{r}'|} d\boldsymbol{r} d\boldsymbol{r}'. \tag{11}$$

Here, the matrix on the left-hand side is the two-center two-electron Coulomb repulsion integral matrix for the auxiliary basis (`int2c2e` in PySCF).

We tested both approaches and found that they yielded comparable performance for accelerating SCF convergence. For all results presented in this paper, we used the first method to generate the ground-truth density coefficients for our training data.

## E METHODS OF CONSTRUCTING HAMILTONIAN MATRIX FROM PREDICTED DENSITY

We outline the construction of the initial Fock matrix for various types of functionals based on the electron density. Specifically, this involves the evaluation of the Coulomb matrix ($\boldsymbol{J}$) and exchange-correlation (XC) matrix ($\boldsymbol{V}_{\text{XC}}$).

The Coulomb matrix is evaluated using three-center electron repulsion integrals:

$$\boldsymbol{J}_{\mu\nu} = \sum_i (\mu\nu|\chi_i)\boldsymbol{c}_i, \tag{12}$$

where

$$(\mu\nu|\chi_i) = \iint \frac{\mu(\boldsymbol{r}_1)\nu(\boldsymbol{r}_1)\chi_i(\boldsymbol{r}_2)}{r_{12}} d\boldsymbol{r}_1 d\boldsymbol{r}_2. \tag{13}$$

are the three-center two-electron integrals between the atomic orbital pair $\mu(\boldsymbol{r})\nu(\boldsymbol{r})$ and the auxiliary function $\chi(\boldsymbol{r})$

**For LDA and GGA functionals**, the electron density and its gradients over the auxiliary basis functions can be readily computed. The XC matrix is then obtained by numerical integration over a set of Becke grids $\boldsymbol{r}_g$ and its weights $\omega_g$:

$$\sum_g \boldsymbol{V}_{xc}[\rho, \nabla\rho]\omega_g\mu(\boldsymbol{r}_g)\nu(\boldsymbol{r}_g). \tag{14}$$

**For meta-GGA functionals**, the XC potential also depends on the kinetic energy density $\tau$. The exact $\tau$ is constructed from molecular orbitals, which are not available when our only input is the total electron density. We therefore approximate $\tau$ using the von Weizsäcker kinetic energy density, which provides an estimate based solely on the density and its gradient:

$$\tau(\boldsymbol{r}) = \frac{1}{2} \sum_i \nabla\psi_i(\boldsymbol{r}) \cdot \nabla\psi_i(\boldsymbol{r}) \approx \frac{\nabla\rho \cdot \nabla\rho}{8\rho}. \tag{15}$$

This allows for the evaluation of the meta-GGA XC matrix term:

$$\frac{1}{2}\sum_g \boldsymbol{V}_{xc}[\rho, \nabla\rho, \tau]\omega_g \nabla\mu(\boldsymbol{r}_g)\cdot\nabla\nu(\boldsymbol{r}_g). \tag{16}$$

**For hybrid and range-separated functionals**, the Hartree-Fock (HF) exchange matrix is needed. The HF exchange term is a function of the density matrix $\boldsymbol{D}$. For a similar reason as with meta-GGA functionals—the difficulty of reconstructing $\boldsymbol{D}$ from $\rho$—we must use an approximate density matrix. We employ the SAD density matrix (i.e. `minao`) as a chemically reasonable guess to construct the HF exchange matrix:

$$\boldsymbol{D}_{\text{SAD}} = \oplus_A \boldsymbol{D}_A, \tag{17}$$

## F  HYPERPARAMETERS AND TRAINING

**Hyperparameters for Model Architectures.**

Table 5: Hyperparameters for model architectures.

| Model | Hyperparameter | Value |
|---|---|---|
| QHNet | radius cutoff | 15.0 |
| | $L_{max}$ | 4 |
| | hidden size | 128 |
| | bottleneck hidden size | 32 |
| | number of layers | 5 |
| | radius embed dim | 16 |
| NequIP-S | radius cutoff | 5.0 |
| | $l_{max}$ | 4 |
| | number of layers | 4 |
| | hidden size | 32 |
| | radial MLP width | 64 |
| NequIP-M | radius cutoff | 5.0 |
| | $l_{max}$ | 4 |
| | number of layers | 7 |
| | hidden size | 64 |
| | radial MLP width | 128 |
| NequIP-L | radius cutoff | 5.0 |
| | $l_{max}$ | 4 |
| | number of layers | 9 |
| | hidden size | 64 |
| | radial MLP width | 128 |

For QHNet models, we adopt the same hyperparameters as those used in Yu et al. (2024). The hyperparameters for the backbone are kept unchanged for different prediction targets to ensure a fair comparison.

For Nequip models, three variants of different sizes are trained and evaluated, namely NequIP-S, NequIP-M and NequIP-L. The sizes of the models are kept the same across different auxiliary basis set choices.

Hyperparameters for all these four architectures are summarized in Table 5.

**Hyperparameters for different prediction targets.** Hyperparameters for training models for different prediction targets are summarized in Table 6. All models have converged after the training finished.

Table 6: Hyperparameters for training.

| Hyperparameter | Hamiltonian | Density Matrix | Density Coefficients |
|---|---|---|---|
| Max Epochs | 5000 | 5000 | 5000* |
| Batch Size | 1024 | 1024 | 1024 |
| Optimizer | Adam | Adam | Adam |
| Learning Rate Scheduler | Polynomial | Polynomial | Polynomial |
| Learning Rate | 5e-3 | 5e-3 | 2e-2 |
| Minimum Learning Rate | 1e-7 | 1e-7 | 1e-7 |

*: The NequIP-S and NequIP-M models are trained for 2000 epochs.

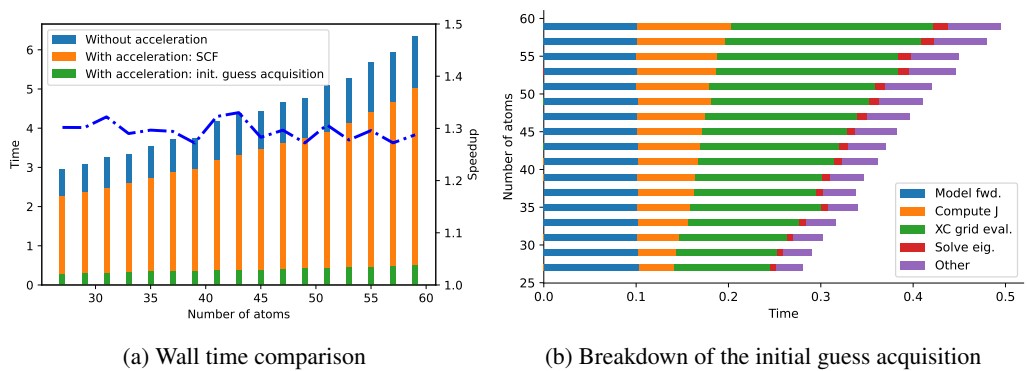

(a) Wall time comparison      (b) Breakdown of the initial guess acquisition

Figure 4: Wall time analysis on the OOD test set. The model we use for acquiring the initial guess is Nequip-L with the def2-universal-jfit basis. To reduce clutter in the figures, we only plot the results for molecules having odd numbers of atoms.

## G    WALL TIME ANALYSIS

While RICs provide a deterministic and hardware-independent measure of initial guess quality, we also provide a wall-clock time analysis to demonstrate practical speedups. However, it is important to note that wall time is heavily implementation-dependent and does not always scale linearly with iteration counts, as different algorithms and molecule sizes shift the computational bottleneck.

We compare the end-to-end wall time required for SCF convergence (inference + initialization + SCF cycles with GPU4PySCF(Wu et al., 2024; Li et al., 2025b)) using our NequIP-L model against the standard `minao` guess on an NVIDIA Tesla V100 GPU. As shown in Figure 4(a), our method achieves a consistent ∼1.3x speedup across system sizes in the OOD test set.

Figure 4(b) breaks down the initialization cost. The procedure to convert predicted coefficients to an initial guess (density matrix) involves three main steps with the following scaling with respect to the number of basis functions $N$:

1. **Coulomb Matrix Construction:** Scales as $O(N^2)$.
2. **XC Grid Evaluation:** Scales as $O(N^2)$.
3. **Eigensolver (First Diagonalization):** Scales as $O(N^3)$.

Although the eigensolver formally scales as $O(N^3)$, for the system sizes tested here, the prefactor is small enough that it is not the bottleneck.

There is significant room for improvement in this overhead. First, the model used is a vanilla NequIP; incorporating state-of-the-art techniques such as SO(2) convolution(Passaro & Zitnick, 2023), FlashTP(Lee et al., 2025) or OpenEquivariance(Bharadwaj et al., 2025) could significantly reduce inference time. Second, the density-to-potential integration is currently implemented in Python; optimizing these kernels would further reduce the pre-SCF overhead.

# H    EXTENDED RESULTS ON SCFBENCH

To provide a more comprehensive view of model performance, we report an extended set of evaluation metrics on SCFbench in Table 7. The definitions of the metrics are detailed below.

**MAE(prediction).** The mean absolute error of the model predictions. Note that values of this metric cannot be compared across different prediction targets.

**MAE($C$).** The mean absolute error of the molecular orbital coefficients obtained from the initial guesses predicted by the model.

**$C$ similarity.** The cosine similarity between the predicted and the ground-truth molecular orbital coefficients.

Table 7: Extended results on the SCFbench dataset. MAE(Prediction) is unitless for density coefficients and density matrix models, but in Hartree for Hamiltonian models. MAE($C$) and $C$ similarity are unitless.

| Prediction Target | Model | ID Test | | | OOD Test | | |
|---|---|---|---|---|---|---|---|
| | | MAE(prediction) ↓ | MAE($C$) ↓ | $C$ Similarity ↑ | MAE(prediction) ↓ | MAE($C$) ↓ | $C$ Similarity ↑ |
| Hamiltonian | QHNet | 4.0e-5 | 0.1527 | 0.8459 | 1.7e-3 | 0.1586 | 0.1572 |
| Density Matrix | QHNet | 5.3e-4 | 0.1314 | 0.9656 | 1.4e-3 | 0.1464 | 0.5101 |
| Density Coefficients def2-universal-jfit | QHNet | 1.7e-4 | 0.0786 | 0.9814 | 5.0e-4 | 0.1085 | 0.8763 |
| | NequIP-S | 2.7e-4 | 0.0995 | 0.9512 | 5.7e-4 | 0.1312 | 0.6863 |
| | NequIP-M | 1.1e-4 | 0.0746 | 0.9846 | 4.0e-4 | 0.0964 | 0.9136 |
| | NequIP-L | 8.9e-5 | 0.0788 | 0.9865 | 3.8e-4 | 0.0928 | 0.9334 |
| Density Coefficients ETB, $\beta = 2.0$ | QHNet | 1.8e-4 | 0.0816 | 0.9815 | 7.0e-4 | 0.1166 | 0.7981 |
| | NequIP-S | 3.5e-4 | 0.1151 | 0.8989 | 8.7e-4 | 0.1486 | 0.4501 |
| | NequIP-M | 1.6e-4 | 0.0832 | 0.9819 | 7.2e-4 | 0.1153 | 0.8113 |
| | NequIP-L | 1.0e-4 | 0.0695 | 0.9907 | 5.4e-4 | 0.0985 | 0.8990 |
| Density Coefficients ETB, $\beta = 1.5$ | QHNet | 1.2e-3 | 0.1096 | 0.9395 | 3.8e-3 | 0.1367 | 0.6119 |
| | NequIP-S | 1.7e-3 | 0.1329 | 0.8298 | 4.2e-3 | 0.1507 | 0.4167 |
| | NequIP-M | 1.2e-3 | 0.1051 | 0.9429 | 4.0e-3 | 0.1375 | 0.5738 |
| | NequIP-L | 8.7e-4 | 0.0887 | 0.9784 | 3.7e-3 | 0.1189 | 0.7767 |

Table 8: WALoss results on the SCFbench benchmark dataset.

| Model | $\xi$ | ID Test | | OOD Test | |
|---|---|---|---|---|---|
| | | Convergence ↑ | RIC ↓ | Convergence ↑ | RIC ↓ |
| QHNet w/o WALoss | - | 100% | 63.20% | 97.43% | 179.47% |
| QHNet w/ WALoss | 1.0 | 100% | 69.56% | 99.57% | 173.07% |
| | 0.5 | 100% | 67.97% | 97.61% | 183.31% |
| | 0.3 | 100% | 67.08% | 98.64% | 179.77% |
| | 0.1 | 100% | 67.11% | 98.40% | 170.57% |
| | 0.01 | 100% | 62.65% | 98.40% | 173.08% |
| | 0.001 | 100% | 62.66% | 97.25% | 177.35% |

# I    RESULTS FOR WAVEFUNCTION ALIGNMENT LOSS (WALOSS)

The Wavefunction Alignment Loss (WALoss) is proposed by Li et al. (2025c) for solving the Scaling-induced MAE-Applicability Divergence (SAD) problem and enhancing the scalability and applicability of Hamiltonian prediction models. As the WALoss is originally used to train the model on the PubChemQH dataset consisting of relatively large molecules, it is thus interesting to find out whether WALoss is able to solve the transferability problem of Hamiltonian prediction. Therefore, although there is no publicly available code for WALoss, we reimplement it ourselves and test it with QHNet on our SCFbench dataset.

There are multiple hyperparameters in WALoss, including the $\lambda_1$, $\lambda_2$ and $\lambda_3$ for weighting the elementwise error losses and WALoss, and the $\rho$ and $\xi$ for weighting the WALoss terms for occupied and unoccupied orbitals, respectively. The optimal values of $\lambda$s are thoroughly discussed in the original paper, but the values of $\rho$ and $\xi$ are unspecified except the description of $\rho \gg \xi$. Therefore,

we adopt the optimal values for the $\lambda$s from the original paper ($\lambda_1 = 1.0$, $\lambda_2 = 1.0$, $\lambda_3 = 2.5$), fix $\rho$ to be 1.0 and experiment with various values for $\xi$.

As listed in Table 8, some settings outperform the original QHNet on the ID test set, but when tested on the OOD test set, none of them show advantage in system size transferability in terms of RIC. We suspect that WALoss is great for solving the learnability problem of the Hamiltonian prediction task on large molecules (such as those in the PubChemQH dataset), but still falls short in overcoming the task's inherent transferability issues.

## J    DIRECT-TO-SOLUTION ACCURACY

If the predicted electron density is sufficiently accurate, one might attempt to perform a single diagonalization to obtain the energy. We evaluate the error of this *direct-to-solution* approach in Table 9.

The results highlight the critical robustness of our density-based approach. On the OOD test set, the Hamiltonian prediction method fails catastrophically, yielding a Mean Absolute Error (MAE) of over 92 Hartree. This magnitude of error aligns with results reported for Hamiltonian models on similar datasets (e.g., WANet without WALoss on PubChemQH (Li et al., 2025c)).

In contrast, our density-based model (NequIP-L) maintains a reasonable error profile ($7.5 \times 10^{-4}$ Hartree MAE) even on OOD systems. This error is already within the well-known chemical accuracy threshold (1 kcal/mol). While the SCF loop is still required to reach precise convergence, our method provides a physically grounded starting point, whereas Hamiltonian-based methods produce unphysical states that require extensive SCF correction.

Table 9: Direct-to-solution results on the SCFbench dataset. All units are Hartree.

| Dataset | Model | MAE($E_{tot}$) ↓ | MAE(HOMO) ↓ | MAE(LUMO) ↑ | MAE(HOMO-LUMO Gap) ↓ |
|---|---|---|---|---|---|
| | QHNet ($H$) | 1.1e-1 | **9.0e-4** | **6.7e-3** | 6.1e-3 |
| **ID Test** | QHNet ($D$) | 3.7e-2 | 7.8e-3 | 7.8e-3 | 2.2e-3 |
| | Nequip-L (jfit) | **1.3e-4** | 1.1e-2 | 1.0e-2 | **1.1e-3** |
| | QHNet ($H$) | 9.2e+1 | 2.4e-1 | 3.4e-1 | 9.5e-2 |
| **OOD Test** | QHNet ($D$) | 4.0e-1 | 6.1e-2 | 6.6e-2 | 2.9e-2 |
| | Nequip-L (jfit) | **7.5e-4** | **8.8e-3** | **8.5e-3** | **2.3e-3** |

## K    RESULTS FOR QHFLOW

Recently, QHFlow (Kim et al., 2025) proposed learning a flow matching generative model for Hamiltonian prediction, achieving state-of-the-art performance on QH9 and MD17. To investigate whether this method mitigates the transferability issue of Hamiltonian prediction, we train a QHFlow model on our SCFbench dataset and test its performance using our standard evaluation pipeline. The hyperparameters we use follow the QH9 defaults from the official QHFlow codebase.

As shown in Table 10, compared with the vanilla QHNet, QHFlow improves the RICs both on the ID and the OOD test sets. However, the significant gap between the ID and OOD performance indicates that the method proposed by QHFlow hardly helps with the inherent transferability issue of Hamiltonian prediction. Nevertheless, the technique could also be applied to the prediction of the density expansion coefficients to obtain an improvement orthogonal to the benefits of our proposed method. We leave this to future work.

Table 10: QHFlow results on the SCFbench dataset.

| Model | ID Test | | OOD Test | |
|---|---|---|---|---|
| | Convergence ↑ | RIC ↓ | Convergence ↑ | RIC ↓ |
| **QHNet** | 100% | 63.20% | 97.43% | 179.47% |
| **QHFlow** | 100% | 56.80% | 98.29% | 146.97% |

## L    LIMITATIONS

While our method demonstrates strong transferability, we acknowledge the following limitations:

- **Chemical Complexity of Large Systems:** While we successfully accelerated systems with up to 900 atoms, our two large-scale cases (polymers and polypeptides) consist of repetitive units. Generalizing to large systems with high chemical diversity remains to be verified.
- **Scope of Systems:** Our evaluation is currently restricted to finite molecular systems; applicability to periodic solids has not yet been validated.
- **Direct-to-Solution Precision:** As noted in Appendix J, our prediction error is not yet low enough to bypass the SCF loop entirely for high-precision applications.

## M    THE USE OF LARGE LANGUAGE MODELS (LLMS)

During the preparation of this work, the authors used LLMs to polish writing and refine sentence structure. The output of LLMs has been reviewed and edited. The authors take full responsibility for all the content in the paper.

