# OpenReview forum: "Towards a Transferable Acceleration Method for Density Functional Theory"
_ICLR.cc/2026/Conference — ICLR 2026 Poster_

### Official Review · Reviewer_arr2 · 2025-10-30

**Soundness:** 3
**Presentation:** 3
**Contribution:** 3
**Rating:** 6
**Confidence:** 3

**Summary:**

The paper introduces a new paradigm for accelerating DFT self-consistent field (SCF) convergence by predicting electron densities in an auxiliary basis using E(3)-equivariant neural networks. Unlike prior models that predict the Hamiltonian (H) or density matrix (D), this approach predicts density coefficients (ρ), from which the Coulomb (J) and exchange–correlation (Vₓc) matrices are reconstructed to form the initial Fock matrix. The authors release SCFbench, a new dataset (43,862 molecules up to 20 atoms, OOD set up to 60 atoms, elements H–S) with Hamiltonians, density matrices, and density coefficients.

**Strengths:**

* Figure1 is interesting. The author demonstrate predicting the electron density could improve the OOD performance. The method transfers across XC functionals (BLYP, SCAN, B3LYP, PBE0) and basis sets (def2-TZVP/QZVP), retaining ≈ 15–25 % speed-up.

* SCFbench is timely; the paper includes careful splits, ID and OOD settings, and extensive reporting (Tables 1–2; Appx. E tables). The dataset spans H, C, N, O, F, P, S with multiple auxiliary bases (def2‑universal‑jfit and ETBs). The summary plots in Fig. 3 (p. 6) are helpful.

**Weaknesses:**

- The predicted electron density $ \\rho $ is not enforced to integrate to the total number of electrons $ N_e $ or remain positive (see App. C).
-§5.1 defines RIC and mentions convergence within 50 iterations, but practitioners need explicit iteration counts and wall-times. Provide a detailed timing breakdown for key components—model forward pass, 3c-ERI contraction for $ \\mathbf{J} $, XC grid evaluation, and eigensolve—on a reference CPU/GPU.
-  The table states “energy unit = Hartree” but includes MAE(C) and C-similarity, which are dimensionless. Clarify the units and definitions used.
-  Please also report the Fock-matrix error $ \\| \\mathbf{F}^{(0)} - \\mathbf{F}^* \\|_F $ and the density error $ \\| \\rho^{(0)} - \\rho^* \\|_2 $, computed on the same grid used for XC integration.
- For meta-GGA, the kinetic energy density $ \tau $ is approximated by the von Weizsäcker form, while for hybrids, HF exchange is built from a superposition-of-atomic-densities density matrix (Appx. C, Eqs. 15–17). Consequently, the reported transferability to SCAN and hybrid functionals assesses a slightly different initial guess than that implied by the predicted $\tilde{\rho}$ alone. This distinction should be explicitly noted when interpreting Table 2 (p. 9).
- Please specify DIIS settings / damping / level shifts used during SCF for all methods; RIC can be sensitive to these solver details (Sec. 5.1)

**Questions:**

See weaknesses.

---

> ### Author Response · Authors · 2025-11-21
> **Response to Reviewer arr2**
>
> We sincerely appreciate the reviewer's constructive feedback on our paper!
>
> ---
>
> **W1:** Predicted density is not enforced to integrate to $N_{e}$ or remain positive.
>
> **Response:** Thank you for your insightful comment. Indeed, the predicted density is not enforced to integrate to $N_{e}$ or remain positive. In fact, this even holds for many of the ground truth density coefficients. However, we would kindly argue that the restriction on the number of electrons or the nonnegativity of density is not a critical requirement for building DFT initial guesses. While unnormalized electrons may introduce errors in certain features, such as charge distributions, total energy estimates, they do not necessarily affect the quality of the SCF initial guess.
>
> We have included a comparison of the electron density with and without constraints in Appendix B: Density Constraints and Normalization. We also analyzed the raw output of our model and found that the number of electron errors hover around 0.1 electrons. Nevertheless, its impact on total energy estimates is minimal (as shown in Appendix J of the revised manuscript).
>
> Here, we would like to highlight two important points:
>
> - The predicted electron density is used to build the initial Fock matrix. The subsequent diagonalization of this matrix selects the lowest $N/2$ orbitals, which inherently enforces the correct number of electrons for the subsequent iteration.
> - Empirically, we found that enforcing normalization on the raw coefficients slightly degraded RIC performance (OOD RIC changes from 66.68% to 67.81%).
>
> ---
>
> **W2:** Comparison of overall wall time.
>
> **Response:** Thank you for your constructive feedback. We agree that both iteration counts and wall times are important to practitioners. We have conducted extra experiments to collect wall time data, and added a dedicated new section G in the Appendix, Wall Time Analysis, to provide the readers with a sense of wall time improvements, where:
>
> - We show a consistent end-to-end wall-clock speedup of $\sim$1.3x on an NVIDIA V100 GPU.
> - We provide a breakdown of the initialization overhead, including model forward pass, 3c-ERI contraction for $\mathbf{J}$, XC grid evaluation and eigensolve. The initialization overhead is nearly constant, and its ratio in total wall time decreases for larger systems.
>
> The following table lists the wall-clock speedups across different atom counts.
>
> | natom      | (20, 30] | (30, 40] | (40, 50] | (50, 60] |
> |------------|----------|----------|----------|----------|
> | speedup    | 1.301    | 1.292    | 1.296    | 1.287    |
>
> The timing breakdown of the initialization overhead is as follows, taking molecules with 60 atoms as an example:
>
> | model_forward | get_j | xc_grid_eval | eigh | other |
> |---------------|-------|-----------|------|--------|
> | 20.2%         | 20.7% | 44.5%     | 3.2% | 11.5% |
>
> For details of the above results, please kindly refer to the newly supplied Appendix G.
>
> Last but not least, as our current code implementation is not fully optimized, we would like to note that there is still potential to further improve the wall time performance.
>
> ---
>
> **W3:** Clarify units and report Fock/density errors.
>
> **Response:** Thank you for your careful review for our manuscript. We have updated the captions of Table 7 and Table 9 to explicitly specify the units of MAE(Prediction), MAE(C), C Similarity, etc. We have also computed the Fock matrix and density error metrics. As the scale of the Frobenius norm varies across the matrix shape, we additionally report the MSE for Fock matrices.
>
> |     |                 | $\|\| \mathbf{F}^{(0)} - \mathbf{F}^* \|\|_F$ | MSE($\mathbf{F}$) | $\|\|\rho^{(0)}-\rho^*\|\|_2$ |
> |-----|-----------------|----------------|--------|-----------------------|
> | ID  | QHNet (H)       | **1.6e-2**         | **1.0e-8** | 3.8e-3                |
> |     | QHNet (D)       | -              | -      | **8.7e-5**                |
> |     | Nequip-L (jfit) | 2.0e-1         | 1.5e-6 | 1.0e-2                |
> | OOD | QHNet (H)       | 4.7            | 3.3e-4 | 5.0                   |
> |     | QHNet (D)       | -              | -      | **6.4e-3**                |
> |     | Nequip-L (jfit) | **2.4e-1**         | **5.9e-7** | 1.2e-2                |

---

> ### Author Response · Authors · 2025-11-21
> **Response to Reviewer arr2 (cont.)**
>
> **W4:** Meta-GGA/Hybrid approximations.
>
> **Response:** We apologize for not making this point clear in the main text. We have added specific text to the Functional and Basis Set Transferability section acknowledging this. We would like to kindly argue that despite this approximation, our method achieves ~85% RIC (15% acceleration) on Hybrid functionals (Table 3). This is a notable result: even with a partial approximation, the high quality of the predicted density-driven terms (Coulomb/XC) provides SOTA acceleration in a zero-shot transfer setting.
>
> ---
>
> **W5:** Specify DIIS settings.
>
> **Response:** We have added a sentence to the Evaluation section specifying: "We use the default PySCF DIIS settings (space size of 8), no level shifting, and no damping."
>
> ---
>
> We sincerely appreciate your encouraging comments on our paper! If there still exist unaddressed concerns, please let us know and we will be more than willing to respond and further improve our paper.

---

### Official Review · Reviewer_LEyV · 2025-10-31

**Soundness:** 1
**Presentation:** 3
**Contribution:** 2
**Rating:** 4
**Confidence:** 4

**Summary:**

This paper proposes a new paradigm for accelerating DFT self-consistent field (SCF) iterations by predicting the electron density coefficients in an auxiliary basis, instead of the Hamiltonian or density matrix. The model employs E(3)-equivariant networks (NequIP and QHNet variants) and is trained on the newly released SCFbench dataset. The approach achieves consistent ∼33% SCF step reduction across systems up to three times larger than training molecules, demonstrating strong transferability across exchange–correlation functionals and basis sets.

**Strengths:**

1. Physically well-motivated formulation; electron density is indeed the fundamental variable of DFT.
2. Public release of SCFbench dataset, beneficial to the community.

**Weaknesses:**

1. Limited practical usefulness:
The proposal does not avoid SCF; it only changes the initial guess. If the model really works, why didn't the authors directly learn the final density of the converged SCF calculations? That will require only one extra step to construct the Hamiltonian to solve for energies or other related physical properties.

2. THe computational saving is unclear:
The paper optimizes “SCF step count,” not wall-clock time including ML inference + integral construction + XC quadrature from the predicted density. Without end-to-end timing on CPU/GPU stacks, it’s hard to judge net throughput gains in realistic workflows.

3. The claim (“universally transferable”) is over-stated:
Evidence is restricted to molecular systems with light elements (H,C,N,O,F,P,S), no periodic solids, and or medium-size molecules with >20 atoms. Transfer tests across functionals/bases are helpful but moderate (e.g., RIC ~85% on B3LYP5/def2-TZVP OOD), which still implies limited speedups in those regimes. True “universality” would need broader chemical domains and periodic boundary conditions.

**Questions:**

1. Wall-clock time: What are the net speedups including ML inference, J and $V_{\mathrm{xc}}$ evaluation from auxiliary density, diagonalization, and SCF mixing—on representative CPU and GPU nodes? Please report time-to-energy for realistic job batches.
2. Direct-to-solution baseline: Have you tried predicting converged density and performing a single update/diagonalization? If not, what prevented it? Empirical evidence would clarify whether your framing is a necessity or a choice.
3. Generalization to slightly larger molecules: WHat is your model performance on larger molecules? For example, MD17/MD22 datasets?

---

> ### Author Response · Authors · 2025-11-21
> **Response to Reviewer LeyV**
>
> We appreciate the reviewer's recognition of the physical motivation behind our method. We have addressed your concerns regarding the "direct prediction" alternative, wall-clock performance, and large-system generalization with substantial new results and text.
>
> ---
>
> **W1 and Q2:** If the model works, why didn't the authors directly learn the final density... and skip SCF?
>
> **Response:**
>
> **Prediction target:** We apologize for the confusion. We have clarified in the text that the model is indeed trained on the final, converged electron density.
>
> **Direct-to-solution:** We thank the reviewer for the constructive advice. To test the model's performance with only one single update/diagnolization, we have conducted additional direct-to-solution experiments by taking the model's predicted electron density and directly deriving various physical properties, including the total energy, the HOMO, the LUMO and the HOMO-LUMO gap.
>
> As shown in our new Appendix J: Direct-to-solution Accuracy, our electron prediction model can provide highly accurate total energy estimates, with errors smaller than 0.1 milli-Hartree in the in-distribution tests and 1 milli-Hartree in the out-of-distribution tests. This level of accuracy is sufficient for direct application in some scenarios. In comparison, the Hamiltonian prediction and density matrix prediction methods produce unphysical energy estimates.
>
> However, this level of accuracy is still not sufficiently reliable for studying out-of-distribution chemical species. Moreover, while direct-to-solution is a valuable direction for ML applications in DFT, we aim to accelerate DFT computations to support a broader range of DFT studies. Various properties, excited states, and studies of many-body effects require the explicit wave function, which can only be reliably derived from DFT SCF computations.  We have added a detailed discussion on this in the new Appendix A: The Role of the SCF Procedure.
>
> ---
>
> **W2 and Q1:** Wall-clock time... need end-to-end timing.
>
> **Response:** We fully agree that knowing end-to-end wall time is important to judge net throughput gains in realistic workflows. Although the RICs reflect the speedup in a hardware or implementation-independent manner, they become meaningless if no wall time improvement is observed. Therefore, we have conducted extra experiments to collect wall time data, and added a dedicated new section G in the Appendix, Wall Time Analysis, to provide the readers with a sense of wall time improvements, where:
>
> - We show a consistent end-to-end wall-clock speedup of $\sim$1.3x on an NVIDIA V100 GPU.
> - We provide a breakdown of the initialization overhead, including ML inference, $\mathbf{J}$, $\mathbf{V}_{\mathrm{xc}}$ evaluation and diagonalization. The initialization overhead is nearly constant, and its ratio in total wall time decreases for larger systems.
>
> The following table lists the wall-clock speedups across different atom counts.
>
> | natom      | (20, 30] | (30, 40] | (40, 50] | (50, 60] |
> |------------|----------|----------|----------|----------|
> | speedup    | 1.301    | 1.292    | 1.296    | 1.287    |
>
> The timing breakdown of the initialization overhead is as follows, taking molecules with 60 atoms as an example:
>
> | model_forward | get_j | grid_eval | eigh | other |
> |---------------|-------|-----------|------|--------|
> | 20.2%         | 20.7% | 44.5%     | 3.2% | 11.5% |
>
> More details of the overall wall-clock speedups and the breakdown can be found in the new Appendix G section.

---

> ### Author Response · Authors · 2025-11-21
> **Response to Reviewer LEyV (cont.)**
>
> **W3 and Q3:** "Universally transferable" is over-stated; evidence restricted to medium-size molecules.
>
> **Response:** We accept the reviewer's point regarding periodic solids, and have changed the title to "Towards a Transferable Acceleration Method..." to be more precise. We have also added a new Limitations section in the Appendix to clarify the scope of our paper.
>
> Regarding molecular size, we have significantly expanded our evaluation. In the revised "Scaling to Large-scale Systems" section, we demonstrate successful transfer to QMugs (100–200 atoms) and Polypropylene chains (905 atoms). Crucially, our model was trained only on molecules with 20 atoms. Transferring to a 900-atom system represents a 45x scale-up without retraining. We respectfully submit that this demonstrates a level of transferability previously unseen in ML-DFT, especially given that baseline methods fail completely on these systems.
>
> |QMugs Dataset| Density Coefficients |             | Hamiltonian |             | Density Matrix |             |
> |---------|----------------------|-------------|-------------|-------------|----------------|-------------|
> | # Atoms | RIC                  | Convergence | RIC         | Convergence | RIC            | Convergence |
> | 100     | 0.75                 | 100%        | 2.24        | 20%         | 1.90           | 50%         |
> | 110     | 0.79                 | 100%        | 2.86        | 30%         | 2.69           | 10%         |
> | 120     | 0.73                 | 100%        | 2.82        | 10%         | 2.33           | 10%         |
> | 130     | 0.78                 | 100%        | -           | 0%          | 3.07           | 10%         |
> | 140     | 0.76                 | 100%        | -           | 0%          | 3.27           | 10%         |
> | 150     | 0.77                 | 100%        | -           | 0%          | -              | 0%          |
> | 160     | 0.79                 | 100%        | -           | 0%          | -              | 0%          |
> | 170     | 0.81                 | 100%        | -           | 0%          | -              | 0%          |
> | 180     | 0.77                 | 100%        | -           | 0%          | -              | 0%          |
> | 190     | 0.82                 | 100%        | -           | 0%          | -              | 0%          |
> | 200     | 0.77                 | 100%        | -           | 0%          | -              | 0%          |
>
> ---
>
> We would like to thank the reviewer for the constructive feedback, and hope the new results address most of the concerns raised! If the reviewer has more concerns, we are happy to supply more evidence and further refine the paper.

---

### Official Review · Reviewer_q3ik · 2025-10-31

**Soundness:** 3
**Presentation:** 3
**Contribution:** 3
**Rating:** 4
**Confidence:** 3

**Summary:**

DFT is one of the most important methods in computational chemistry, offering the possibility of computing energies and forces for (weakly correlated) molecules and solids with high accuracy and, compared to other methods, modest computational effort. DFT calculates these quantities from the electron density using auxiliary (and empirical) potential functions. Finding the electron density requires the solution of a nonlinear eigenvalue problem, which can only be tackled using iterative algorithms (the most famous among them is called SCF). Such iterative algorithms crucially hinge on good choices of initial values, within the basis of attraction of the SCF dynamics. The paper introduces an ML-based method to obtain such initial values.

Contrary to previous approaches, the proposed method directly operates on expansion coefficients of the electron density with respect to a given basis. From this, the matrices which make up the nonlinear eigenvalue problem can be computed via quadrature. This representation of the electron density has the advantage of being transferable to larger systems. Indeed it is shown that training on systems with up to 20 atoms yields equally good initializations for systems with up to 60 atoms.

**Strengths:**

The idea of directly using the electron density allows to obtain a transferable initialization method is very nice. In particular the possibility to train on small systems while still getting good initializations on larger systems is extremely promising.

**Weaknesses:**

The claim of "universal transferability" might be a bit strong given that the method is only tested on systems with up to 60 atoms - keeping in mind that conventional DFT calculations often treat systems with hundreds or thousands of atoms. More evidence on larger scales would be needed to fully validate this claim.

Unless I am missing something the gain in the number of iterations is around 34% for a certain experiment but varies considerably across different experiments (see Table 2).

The paper seems to exclusively focus on RIC. However, this is not the only indicator of efficiency. In order to make a fair comparison one needs to monitor the wall time (keeping in mind that assembling the relevant matrices from the expansion coefficients requires the numerical solution of complicated quadrature problems) as well as the final accuracy. Without a careful comparison of overall wall time and accuracy the results become meaningless because we do not know the true complexity, or what the algorithm converges to. Therefore, this information has to be included in a revision.

**Questions:**

A detailed comparison in wall time and accuracy has to be included.

The authors might reconsider the term "universal transferability", as it might be a slight overstatement.

---

> ### Author Response · Authors · 2025-11-21
> **Response to Reviewer q3ik**
>
> We thank the reviewer for their positive assessment of our work's soundness and the potential of our density-based approach. We have extensively revised the manuscript to address these points.
>
> ---
>
> **W1 and Q2:** The claim of "universal transferability" might be a bit strong... More evidence on larger scales would be needed.
>
> **Response:** We agree with the reviewer that "universal" is a bit strong given that our evaluation does not currently cover periodic solids. We have added a dedicated "Limitations" section (Appendix K) to explicitly outline the current scope (finite molecular systems) and the need for future validation on periodic boundary conditions. We have also changed the title to "Towards a Transferable Acceleration Method for Density Functional Theory".
>
> To demonstrate the capability of the proposed initial guess method on large-scale systems, we have added a new section "Scaling to Large-scale Systems" with results that significantly exceed the original 60-atom tests, including:
>
> 1. QMugs dataset: Molecules with 100–200 atoms.
> 2. Macromolecules: A Glycine-100 polypeptide (703 atoms, converged in 10 iterations (vs. 17 for minao)) and a Polypropylene chain (905 atoms,  converged in 8 iterations (vs. 12 for minao)).
>
> |QMugs Dataset| Density Coefficients |             | Hamiltonian |             | Density Matrix |             |
> |---------|----------------------|-------------|-------------|-------------|----------------|-------------|
> | # Atoms | RIC                  | Convergence | RIC         | Convergence | RIC            | Convergence |
> | 100     | 0.75                 | 100%        | 2.24        | 20%         | 1.90           | 50%         |
> | 110     | 0.79                 | 100%        | 2.86        | 30%         | 2.69           | 10%         |
> | 120     | 0.73                 | 100%        | 2.82        | 10%         | 2.33           | 10%         |
> | 130     | 0.78                 | 100%        | -           | 0%          | 3.07           | 10%         |
> | 140     | 0.76                 | 100%        | -           | 0%          | 3.27           | 10%         |
> | 150     | 0.77                 | 100%        | -           | 0%          | -              | 0%          |
> | 160     | 0.79                 | 100%        | -           | 0%          | -              | 0%          |
> | 170     | 0.81                 | 100%        | -           | 0%          | -              | 0%          |
> | 180     | 0.77                 | 100%        | -           | 0%          | -              | 0%          |
> | 190     | 0.82                 | 100%        | -           | 0%          | -              | 0%          |
> | 200     | 0.77                 | 100%        | -           | 0%          | -              | 0%          |
>
> **Result:** Our model, which was trained exclusively on molecules with <20 atoms, successfully converged these systems (up to 900 atoms) with consistent acceleration. In stark contrast, Hamiltonian and density matrix prediction methods failed to converge for medium size systems and ran out of memory with large molecules. We believe this 45x extrapolation in system size is strong evidence of the method's robustness.
>
> ---
>
> **W2:** The gain in iterations is around 34%... varies considerably.
>
> **Response:** We would like to kindly argue for this point. While a 33% reduction might appear modest in isolation, it represents the State-of-the-Art (SOTA) in this domain. For example, previous SOTA methods (e.g., WANet with WALoss [1]) achieve ~85% RICs but rely on in-distribution training (large molecules, same functionals and basis sets). In contrast, our method achieves this comparable performance zero-shot - trained only on small (<20 atoms) PBE/def2-SVP data and transferring to completely different functionals and much larger basis sets. This demonstrates genuine physical transferability.
>
> [1] Li, Yunyang, et al. "Enhancing the Scalability and Applicability of Kohn-Sham Hamiltonians for Molecular Systems" ICLR 2025

---

> ### Author Response · Authors · 2025-11-21
> **Response to Reviewer q3ik (cont.)**
>
> **W3 and Q1:** Comparison of overall wall time... without this, results become meaningless.
>
> **Response:** Thank you for your constructive advice. Indeed, although the RICs reflect the speedup in a hardware-independent manner, they become meaningless if no wall time improvement is observed. Therefore, we have conducted extra experiments to collect wall time data, and added a dedicated new section G in the Appendix, Wall Time Analysis, to provide the readers with a sense of wall time improvements, where:
>
> - We show a consistent end-to-end wall-clock speedup of $\sim$1.3x on an NVIDIA V100 GPU.
> - We provide a breakdown of the initialization overhead. The initialization overhead is nearly constant, and its ratio in total wall time decreases for larger systems.
>
> The following table lists the wall-clock speedups across different atom counts. Please refer to Appendix G for more details.
>
> | natom      | (20, 30] | (30, 40] | (40, 50] | (50, 60] |
> |------------|----------|----------|----------|----------|
> | speedup    | 1.301    | 1.292    | 1.296    | 1.287    |
>
> As for the accuracy, we verify on the OOD test set that the accelerated calculations and the baseline converge to the exact same energy (within the convergence threshold of $10^{-10}$ Hartree), ensuring correctness.
>
> ---
>
> Thank you for the insightful comments! We sincerely hope these new results and clarifications help resolve most of your concerns.

---

### Author Response · Authors · 2025-12-03
**Review Summary by Authors**

Dear AC,

Thank you for your careful consideration of our manuscript. We have revised the manuscript to address all concerns and questions raised during the review process. To facilitate your assessment, we summarize the main points of the feedback and our responses below:

**1. Demonstrated "Zero-Shot" Scalability & Revised Scope**

To address concerns about "universality" (Reviewers q3ik, LEyV), we revised the paper title to *"Towards a Transferable Acceleration Method..."* and added a new section (*Scaling to Large-scale Systems*) demonstrating that our model, which was trained **only** on molecules with <=20 atoms, successfully accelerates:

* Polypropylene polymer (905 atoms) and Glycine-100 (703 atoms).
* QMugs (100–200 atoms) with 100% convergence.

This represents a **45x extrapolation** in system size where baseline Hamiltonian methods failed to converge or ran out of memory.

**2. State-of-the-Art Transferability**

Regarding the magnitude of acceleration (Reviewer q3ik): While our best-case reduction is \~34% (66% RIC), our method maintains **~85% RIC** even in the most challenging **zero-shot transfer** scenarios (e.g., transferring to Hybrid B3LYP5 functionals and def2-TZVP basis sets). This matches the performance of current SOTA methods (e.g., WANet w/ WALoss) which typically require **in-distribution** training to achieve that same 85%. This proves our method provides SOTA-level acceleration on unseen physics without the need for retraining.

**3. Verified Wall-Clock Speedup**

To demonstrate practical utility beyond iteration counts, we include end-to-end wall-clock benchmarks (Appendix G). These results show a consistent ~1.3× speedup on an NVIDIA V100 GPU, along with favorable scaling of initialization overhead.

---

We sincerely thank the ACs for their oversight and the ICLR community for the opportunity to present our work. We also express our deep appreciation to the reviewers for their thoughtful comments, which have substantially strengthened the manuscript. By releasing SCFbench and demonstrating the method’s applicability and transferability, we hope this work contributes meaningfully to ongoing research in this area.

Best Regards,

Authors

---

### Meta-Review · Area_Chair_TPkj · 2025-12-14

**Summary:**

This paper proposes a method to accelerate DFT calculations through providing an initial guess of electron density, which in result reducing the number of SCF iterations. The method is simple, building upon existing neural network architectures like NequIP and QHNet. In the experiments, The authors convincingly show how the proposed method consistently accelerate DFT calculations.

Reviewers were mainly concerned about exaggerated claims, lack of wall clock comparison, and limited practical usefulness. After reading the rebuttal thoroughly, I believe the authors' rebuttal successfully clarified all the concerns.

Overall, I would like to suggest acceptance for this work, since there remains no significant concerns of the reviewers.

However, if the paper gets accepted, I strongly urge the reviewers to add additional baselines in their final manuscript. The reviewers only compare with QHNet, but QHFlow seems to be the current SoTA models with significant performance improvements for Hamiltonian prediction. Without the comparison, the claim of the proposed method being "the first candidate for transferable DFT acceleration" does not hold.

**Reviewer Concerns:**

Exaggerated claim for universal transferability (q3ik, LEyV): The authors renamed the paper to avoid this claim.

Lack of wall clock comparison (q3ik, LEyV): The authors provided additional wall clock comparison.

Limited practical usefulness due to unavoidable SCF iterations (LEyV): The authors clarified misunderstanding of the reviewer.

**Reviewer Scores:**

Reviewer q3ik and LEyV would have changed their score, since their concerns are directly resolved.

---

> ### Public Comment · ~Siyuan_Liu3 · 2026-03-01
> **Response to Meta Review**
>
> Dear Area Chair,
>
> Thank you for your positive assessment and constructive suggestion regarding stronger baselines.
>
> Following your recommendation, we have added a new appendix section evaluating QHFlow[1] within our SCF acceleration framework. Specifically, we trained QHFlow on our SCFbench dataset using the official QH9 default hyperparameters and evaluated it under our standard pipeline. The results are reported in Appendix K, Table 10.
>
> We observe that QHFlow improves the RIC metrics over vanilla QHNet on both ID and OOD test sets. However, a substantial performance gap between ID and OOD remains, with the average RIC on the OOD test set still above 100%, indicating that QHFlow does not fundamentally resolve the inherent transferability limitation of Hamiltonian prediction. Consequently, while stronger Hamiltonian models improve in-distribution accuracy, they do not alter the core transferability challenge addressed by our work.
> Importantly, these results further support our positioning: our contribution is a general SCF acceleration framework whose effectiveness is largely orthogonal to advances in Hamiltonian prediction architectures. Improvements in predictors such as QHFlow can be directly incorporated into our pipeline, and may also benefit other predicted physical quantities (e.g., density expansion coefficients), which we leave for future work.
>
> Thank you again for helping us strengthen the final version of the paper!
>
>
> Sincerely,
>
> Authors
>
> [1] Kim et al. High-order Equivariant Flow Matching for Density Functional Theory Hamiltonian Prediction, NeurIPS 2025.

---

### Decision · Program_Chairs · 2026-01-26

Accept (Poster)